# The role of Methyl-CpG binding domain 3 (Mbd3) protein in epileptogenesis

Karolina Nizinska[1], Maciej Olszewski[1], Sandra Binias[2], Dorota Nowicka[1], Kinga Szydlowska[1], Kinga Nazaruk[1], Bartosz Wojtas[2], Katarzyna Lukasiuk[1]*

1 Laboratory of Epileptogenesis, Nencki Institute of Experimental Biology, Warsaw, Poland, 2 Laboratory of Sequencing, Nencki Institute of Experimental Biology, Warsaw, Poland

* k.lukasiuk@nencki.edu.pl

## Abstract

Methyl CpG binding domain 3 (Mbd3) protein belongs to the MBD family of proteins and is responsible for reading the DNA methylation pattern. Our previous study revealed increased levels of Nucleosome Remodeling and Deacetylase (NuRD) complex proteins, including Mbd3, in the brains of epileptic animals. The present study investigated whether the Mbd3 protein level determines the seizure threshold. We demonstrate that seizures induced by pentylenetetrazole (PTZ) cause a transient, brain area-specific increase in Mbd3 protein levels in the entorhinal cortex and amygdala. Overexpression of Mbd3 in the amygdala using AAV decreased anxiety, increased excitability in the open-field test, and accelerated epileptogenesis in the PTZ-kindling model. In vitro, mRNA profiling using RNA-seq in a model of magnesium deficiency-induced epileptiform discharges revealed complex, time- and state-specific changes in gene expression. Genes regulated by Mbd3 overexpression were associated with the Wnt and Notch pathways, potassium channel function, and GABAB receptor signaling. Our findings indicate that increased Mbd3 expression has pro-epileptic properties and contributes to the regulation of multiple pathways potentially involved in seizure development. Significantly, seizures themselves transiently elevate Mbd3 levels, suggesting a potential vicious circle that may aggravate disease progression. Targeting the pro-epileptic effects of Mbd3 could therefore represent a novel therapeutic approach in epilepsy.

## Introduction

Epilepsy is one of the most common neurological diseases characterized by long-term brain dysfunction manifested by spontaneous seizures and neurobiological, cognitive, psychological, and social consequences [1–5]. Increasing evidence suggests that DNA methylation may play an essential role in the development of epilepsy [6–9]. For example, increased levels of DNA methyltransferases DNMT1 and DNMT3a were observed in the temporal cortex of patients with temporal lobe epilepsy (TLE)

**Data availability statement:** sequencing data are deposited at NCBI Gene Expression Omnibus (https://www.ncbi.nlm.nih.gov/geo/query/acc.cgi?acc=GSE227084).

**Funding:** This work was supported by an OPUS grant 2015/19/B/NZ4/01401 (KL, KN). There was no additional external funding received for this study.

**Competing interests:** The authors have declared that no competing interests exist.

[10]. In contrast, another study revealed a decrease in DNMT3a2 isoform expression in the hippocampus of TLE patients and a decrease in DNMT3a1 in the hippocampus of TLE patients with a history of FS. In the neocortex, the expression of DNMT1 and DNMT3a1 was increased in TLE patients [11]. Decreased levels of 5-methylcytosine and 5-hydroxymethylcytosine, markers of global DNA methylation, have been observed in the hippocampus of TLE patients with a history of febrile seizures [11]. In contrast, increased levels of 5-methylcytosine have been found in the neocortex of patients with TLE [11].

Changes in DNA methylation patterns differ between animal models of epilepsy and occur in gene promoters, introns, exons, and non-coding regions [6]. The issue of whether changes in methylation contribute to the development of epilepsy, affect the disease process, or are a reaction to seizures is an open question. At least for inflammation-related genes in the hippocampus of mesial temporal lobe epilepsy with hippocampal sclerosis (MTLE-HS) patients, methylation increases with the duration of the disease, which can be explained by disease progression or the increased cumulative exposure to seizures [12]. The dietary interventions that influence DNA methylation levels, such as a maternal methyl-enriched diet for controlling audiogenic or absence seizures in rats, or a ketogenic diet, suggest a role of DNA methylation in epileptogenesis and seizure expression [13–18].

Methyl CpG binding domain 3 (Mbd3) protein belongs to the MBD family of proteins that bind to 5-methylcytosine through their conserved methyl-CpG binding domain (MBD) and mediate transcriptional silencing of methylated promoters by attaching a silencing complex [19,20]. The MBD domain of the Mbd3 protein does not selectively recognize methyl-CpG islands. It can bind to 5-hydroxymethylcytosine and unmethylated DNA [21,22]. Little is known about the involvement of the Mbd3 protein in epilepsy development [23–25]. The study by Wang et al. showed that lowering Mbd3 expression with shRNA in mice leads to a reduction in the intensity of pilocarpine-induced seizures, decreasing the proportion of mice entering Racine stage IV seizures from 80% to 0% [25]. Our previous studies, conducted by Bednarczyk et al., showed that Mbd3 mRNA is present in neurons, oligodendrocytes, and astrocytes in both control and epileptic rats in the amygdala stimulation model of epilepsy throughout the brain [23]. Mbd3 protein is also abundant in the brain and has been detected in the nuclei of neurons, mature oligodendrocytes, and a subpopulation of astrocytes, but not in microglia. We identified increased Mbd3 protein levels in the amygdala and piriform cortex of animals with epilepsy [23]. Moreover, using ChIP-Seq, we identified alterations in the DNA binding of the Mbd3-containing NurD complex in experimental animals compared with the sham group [23]. For several genes, changes in Mbd3 binding to their promoters occurred only in rats with spontaneous seizures [23]. This suggests that Mbd3 regulates gene expression during epileptogenesis, epilepsy, or seizure induction.

In the present study, we aimed to elucidate whether elevating the Mbd3 protein level affects seizure threshold in acute seizures induced by pentylenetetrazole (PTZ) and epileptogenesis in the PTZ kindling model. Moreover, to elucidate the molecular mechanisms of Mbd3 function, we characterized alterations in gene expression

resulting from increased Mbd3 levels in an *in vitro* model of epileptiform discharges. Our data suggest that the Mbd3 protein contributes to the development of epilepsy.

## Materials and methods

### Animal

All of the animal procedures were approved by the Ethical Committee (permits no. 357/2017, 395/2017, and 838/2019) of the Warsaw Local Ethics Committee for Animal Experimentation and conducted according to guidelines established by the European Council Directive 2010/63/EU and the ARRIVE guidelines [26].

Male Sprague-Dawley rats (250–270 g) from the Mossakowski Medical Research Centre, Polish Academy of Sciences (Warsaw, Poland), were used in this study. The rats were housed under controlled conditions (24°C, 50–60% humidity, 12 h light/12 h dark cycle) with food and water available *ad libitum*. The animals were housed in pairs in an enriched environment. Altogether, 220 animals were used; two animals died during PTZ kindling. The body weight at the start of the experiment was 280–320 g.

### Animal surgery

Electrode implantations were performed according to the procedure described by Nissinen et al. 54, with modifications introduced by Guzik-Kornacka et al. [27]. Animals that reached a weight of 300 g underwent surgery under isoflurane anesthesia (Bartex, initial dose of 4%, then maintained at 1.5% to 2% in oxygen) and received analgesia with 0.2 mg butorphanol (Butomidor, 10 mg/ml, Richter Pharma AG). The EEG surface electrode for seizure monitoring was implanted stereotactically over the frontal cortex (AP: 3.0; L: +2.0 mm from Bregma, #E363/20, PlasticOne). The reference and ground electrodes were placed over the cerebellum (AP: 10.0 mm; L: ±2.0 mm from Bregma, #E363/20, PlasticOne). The ends of the electrodes were inserted into a socket (#E363/2-TW/Spec, PlasticOne) and secured to the skull using dental acrylic (Duracryl Plus, SpofaDental).

The adeno-associated viruses (AAV) were designed and ordered from Tebu-bio (39 rue de Houdan, 78610 Le Perray-en-Yvelines, France). To increase the Mbd3 expression level, AAV-SYN-Mbd3-GFP was used. AAV stocks were diluted 1:10 in sterile PBS buffer before injection into the rat's brain. AAV was injected bilaterally into the BLA. The injection site coordinates were: AP: −2.8; L: ±4.7; DV: −7.2 59. AAVs were injected in a volume of 0.4 µl per hemisphere (0.8 µl per rat) at a rate of 0.2 µl per minute.

### Behavioral tests

In the behavioral hyperexcitability test, animal behavior was assessed in four categories: approach response, touch response, loud noise response, and pick-up response, as previously described [28,29]. In the approach-response test, a pen held vertically was slowly moved toward the animal's face. Responses were scored as follows: 1- no reaction; 2 – sniffing the pen; 3 – moving away from the pen; 4 – freezing; 5 – jumping out; and 6 – attacking the pen. In the touch-response test, the animal was gently prodded in the rump with the blunt end of the pen. Responses were scored as follows: 1 – no reaction; 2 – turning toward the touched area; 3 – moving forward, away from the touch; 4 – freezing; 5 – jerking toward the touch; 6 – turning away from the touch; and 7 – jumping with or without vocalization. In the loud noise test, a clicking sound was generated by a timer several centimeters above the animal's head. Responses were scored as follows: 1 – no reaction; 2 – jumping slightly, flinching, or flicking the ears; and 3 – jumping abruptly. In the pick-up test, the animal was picked up by grasping around its body. Responses were scored as follows: 1 – very easy; 2 – easy with vocalization; 3 – some difficulty, with the rat rearing and facing the experimenter's hand; 4 – freezing with or without vocalization; 5 – difficult, with the rat avoiding the hand; and 6 – very difficult, with the rat behaving defensively, with or without attacking the experimenter's hand. The behavioral hyperexcitability test was repeated four times throughout one day, at one-hour intervals; Median scores were used for subsequent data analysis.

The open-field test was conducted in a square, dark gray box measuring 1 m x 1 m with walls 35 cm high. This arena was constructed by the Laboratory of Animal Models at the Nencki Institute of Experimental Biology. All tests were recorded using a Samsung SCB-2000P and the WinTV program (Hauppauge, USA). During the open-field test, the rat was placed in the center of the arena, and its behavior was monitored using WinTV software (Hauppauge, USA) for 15 minutes. The camera was positioned just above the arena in the middle section. Files were analyzed using EthoVisionXT software (version 8.5, Noldus, USA), and the rat's movements were tracked for the same duration. The latency to enter the inner area of the arena, the latency to enter the central area, and the speed were recorded during the test.

The elevated cross maze comprised two open arms (50 cm long, 14 cm wide) and two closed arms (with a wall height of 29 cm), elevated 50 cm above the floor. A rat was placed in the central area of the maze and monitored for 15 minutes. The number of entries into closed and open arms, as well as the speed of movement, were recorded during the test.

The open-field and elevated plus-maze tests were recorded using WinTV software (Hauppauge, NY, USA). Video files were analyzed using EthoVision 8.5 software (Noldus, Leesburg, VA, USA), and the data were exported to Microsoft Excel. The order of animals within trials was randomized.

## PTZ challenge

PTZ (pentylenetetrazole) dissolved in saline was applied intraperitoneally at a convulsive dose (50 mg/kg body weight) [30,31]. Control animals received an intraperitoneal injection of saline. Rats were monitored with video-EEG (Panasonic WV-CP480; Comet EEG system, Grass Technologies, USA) for 60 minutes after PTZ injection. EEG recordings were analyzed manually to detect the start of electrographic seizures using TWin EEG software (v.4.5.3.23, Grass Technologies, USA). Video was used to detect behavioral generalized tonic-clonic seizures. Animals for mRNA analysis were sacrificed at 1, 4, 8, 24, and 48 h after the seizure initiation; animals for protein analysis were sacrificed at 4, 8, 24, and 48 h after the seizure initiation. The mortality rate during the PTZ challenge was 0%.

## PTZ kindling

During PTZ kindling, rats received intraperitoneal injections of PTZ at a dose of 35 mg/kg, 3 times per week [32,33]. This dose of PTZ in our hand is routinely used for kindling. In the present cohort, some animals had low-grade seizures already during the first session. Behavioral seizures were recorded for 30 minutes using a video camera (Panasonic WV-CP480) and scored according to the Racine scale [34]. The criterion for full kindling was defined as the induction of seizure scores 4–5 according to the Racine scale during three consecutive sessions. The experiment was conducted until all animals met the accepted criterion of the kindling model. For ethical reasons, each animal reaching the criterion (3 consecutive sessions with tonic-clonic convulsions) was withdrawn from the experiment and did not participate in subsequent sessions. The mortality rate was 10% (1 animal per group).

## Epileptic discharges *in vitro*

Embryos at 18 days post-fertilization (E18) were used to establish primary cultures of cortical neurons, as described by Xu et al. [35]. The extracted cortex was trypsinized at 37°C for 15 minutes in 0.2% Trypsin (#27250−0180, ThermoFisher) and 0.15 mg/ml DNase (#DN-25, Sigma-Aldrich) in HBSS. Trypsinization was stopped with 10% FBS (#10106−151, ThermoFisher) diluted in HBSS. Two hundred thousand cells per well were seeded into a 12-well plate pre-coated with poly-D-lysine (five µg/ml) in 0.1 M borate buffer (#P7280, Sigma-Aldrich). Cultures were grown in an incubator at 37°C and 5% $CO_2$. On days 2 and 6, fifty percent of the medium was exchanged for FBS serum-free medium (1x B-27, 10 mg/ml Gentamicin, 0.5 mM Glutamax in Neurobasal Medium). Cells were transfected on the 9DIV with 1µl of AAV suspension.

Epileptiform discharges *in vitro* were induced as described by Jiang et al. with some modification [36]. Briefly, cultures were incubated for 3 hours in pBRS buffer without magnesium (145 mM NaCl, 2.5 mM KCl, 10 mM HEPES, 2 mM $CaCl_2$, 10 mM glucose, 0.002 mM glycine, pH=7.3). The control cultures were incubated in pBRS buffer with magnesium (145 mM

NaCl, 2.5 mM KCl, 10 mM HEPES, 2 mM CaCl2, 10 mM glucose, 0.002 mM glycine, 1 mM MgCl2, pH=7.3). At the end of the incubation, the pBRS buffer was replaced with culture medium. Cell viability was determined using a colorimetric MTT assay [37].

## RNA isolation

Animals were anesthetized with isoflurane and decapitated. Tissue from the left hemisphere was stored in StayRNA buffer (#038–500, A&A Biotechnology) for mRNA isolation. The RNA isolation was performed using the RNeasy Mini Kit (#74104, Qiagen) according to the manufacturer's recommendations. RNA concentration was measured using a Nano-drop Spectrophotometer (DS-11 Spectrophotometer, DeNovix) at wavelengths of 260 nm and 280 nm. The isolated mRNA was stored at −80°C.

RNA from the cell culture was isolated using Qiazole (#79306, Qiagen), followed by purification with chloroform (#C0549-1PT, Sigma) and the RNA isolation kit: RNeasy Mini Kit (#74104, Qiagen). The RNA was measured using a Nanodrop Spectrophotometer (DS-11 Spectrophotometer, DeNovix) at wavelengths of 260 nm and 280 nm. The isolated RNA was stored at −80°C.

## PCR

Real-time PCR reactions were performed using the Fast SYBR™ Green Master Mix kit (#4385612, Life Technologies). Samples were prepared according to the protocol included in the reaction kit, followed by the addition of primers: Mbd3 (sequence from 5' to 3', forward: TTACTATAGCCCCAGCGGGA, reverse: AGTGCAGTGTTCAGGTCAGG), or Gapdh (sequence from 5' to 3', forward: GATTTGGCCGTATCGGAC, reverse: GAAGACGCCAGTAGACTC).

## Protein isolation and western blot

Animals were anesthetized with isoflurane and decapitated. Tissue from the right hemisphere was used for protein isolation. Tissue was homogenized using a Tissue Ruptor homogenizer (Qiagen) in the lysis buffer containing 0.5% Triton X-100, 0.5 M KCl, 0.5 M PIPES, 1 M MgCl2, 0.5 M EGTA, 1x concentrated protease inhibitor cocktail, 100 mM PMSF, and 1 M DTT. The homogenate was incubated frozen, thawed on ice, and centrifuged (4°C for 20 minutes at 11,000 rpm). The supernatant was centrifuged again (4°C, 20 minutes, 11,000 rpm). The supernatant was stored at −80°C. Protein concentration was determined using the Bradford method.

Cells *in vitro* were scraped in the lysis buffer containing 0.5% Triton X-100, 0.5 M KCL, 0.5 M PIPES, 1 M MgCl2, 0.5 M EGTA, 1x concentrated protease inhibitor cocktail, 100 mM PMSF, 1 M DTT, frozen, thawed, and centrifuged (four°C, 20 minutes at 11,000 rpm). The Supernatant was centrifuged again (4°C, 20 minutes, 11,000 rpm). Next, the supernatant was at −80°C. Protein concentration was determined using the Bradford method.

For Western blot, protein samples were subjected to Tris-glycine sodium dodecyl sulfate- polyacrylamide gel electrophoresis (SDS-PAGE) and transferred to the nitrocellulose membranes (#RPN82D, Amersham Hybond-ECL). For immunodetection, nonspecific binding was blocked by incubation in 5% non-fat milk in TBS-T (0.5 M Tris, 0.9% NaCl, 0.1% Tween 20, pH 8.0). Blots were then probed with primary polyclonal antibody anti-MBD (rabbit, #A302-528A, Bethyl Laboratories, concentration of 1:1000 in TBS-T), followed by secondary antibody (goat anti-rabbit conjugated with horseradish peroxidase antibody, AP132P, Merck Millipore, concentration of 1:20,000 in TBS-T). According to the manufacturer's instructions, chemiluminescent signals were detected with the ECL Western Blotting Analysis System (GERPN2106, Amersham). For stripping, a 2% SDS solution in 90 mM glycine (pH 3) was used. Next, membranes were incubated for 60 minutes with 5% nonfat milk in TBS-T and for 2 hours with an antibody against β-actin (Horseradish peroxidase-conjugated, mouse monoclonal antibody, #A3854, Sigma-Aldrich, concentration of 1:20,000 in TBS-T). Signals were detected by chemiluminescence with the ECL Western Blotting Analysis System.

## Immunofluorescence and image analysis

Rats were anesthetized with isoflurane, followed by an intraperitoneal injection of 2 mL/kg pentobarbital (Morbital, Biowet, 133.3 mg/mL). Perfusion was performed with 200 ml of saline, followed by 200 ml of 4% paraformaldehyde in PBS (pH 7.4). The brains were post-fixed in a 4% PFA solution for 4 hours and then cryoprotected in 30% sucrose in 0.02 M KPB (pH 7.4) solution at 4°C. The brains were frozen on dry ice and stored at −80°C. They were then cryosectioned into 30 μm sections in a 1-in-5 series in the coronal plane using a cryostat (Leica CM1860). Sections were stored in TCS buffer (30% ethylene glycol, 25% glycerol, 0.05 M PB buffer) at −20°C.

Sections were stained with primary monoclonal antibody (chicken anti-Neun, #ABN91, Sigma-Aldrich/Merck or mouse anti-GFAP, #MAB3402, Millipore, or mouse anti-GFAP, # MAB3402, Sigma-Aldrich/Merck, 1:1000) followed by anti-chicken Alexa Fluor 568 (#A11041 ThermoFisher Scientific, 1:2000) or horse anti-mouse Texas Red™ (#TI-2000, Vector Laboratories, 1:2000), respectively. Sections counterstained with Hoechst (#62249, ThermoFisher 1:1000), mounted, and coverslipped in Vectashield® Mounting Medium (#H-1000, Vector Laboratories).

The sections were photographed using a Nikon Eclipse 80 microscope and a Lumen 200 fluorescent lamp (Prior Scientific), with a Nikon 10/0.30 DIC L/N1 objective. Images covering the virus expression were superimposed to visualize the co-localization of GFP with NeuN and Hoechst staining using ImagePro Plus 5.0.

To analyze the efficacy of the *in vivo* transfection, the numbers of GFP+ cells, NeuN+ neurons, and double-stained cells were assessed.

For the measurements of virus spread in the brain tissue, the volume within which the GFP+ appeared was calculated. To this end, the area containing GFP+ was measured in every fifth section (30 μm). The volume was calculated by multiplying these areas by the distance between the sections and summing the results. Fourteen to sixteen sections covering the whole extent of virus expression were used for calculations.

To evaluate the transfection efficacy in neuronal cultures, 24–29 non-overlapping images were captured along the axis of the culture round coverslip, using different filters for GFP, AlexaFluor 568/Texas Red, and Hoechst, with a Nikon 10x/0.30 DIC L/N1 objective. The numbers of GFP+ cells, NeuN+ neurons, and double-stained NeuN+/GFP+ neurons were assessed. The percentage of NeuN+ neurons containing GFP was calculated.

For the presentation in the figures, images were acquired using a Zeiss LSM780 confocal microscope equipped with a Plan-Apochromat 63×/1.4 NA oil immersion DIC objective. Figures were prepared using Adobe Photoshop 7.0 and Corel-DRAW 12. Brightness and contrast were adjusted to regain the sections' natural appearance.

## RNA sequencing

RNA-seq libraries were prepared with the KAPA Stranded mRNA Sample Preparation Kit according to the manufacturer's protocol (Kapa Biosystems, MA, USA) as described [38]. Transcriptomic data analysis was performed: FASTQ files were aligned to the rn6 rat reference genome using the STAR program [39], and reads were counted against genes using the featureCounts algorithm [39]. Gene counts were normalized with the FPKM method, and differential analysis was performed by DESeq2 [40]. Genes were considered to be differentially expressed (DE) with an adjusted p-value < 0.05 [41]. Cluster analysis was performed using R software and the Mfuzz Clustering package. Functional analysis of gene groups was performed using the David database (https://david.ncifcrf.gov/) accessed in August 2025 [42,43].

## Statistical analysis

Normality of data distribution was assessed using the Kolmogorov–Smirnov test. The non-parametric Mann-Whitney test was used to investigate the differences in mRNA and protein levels between control animals and the PTZ-treated group, the latency of electrographic seizures in the PTZ challenge, group comparisons at each session in PTZ kindling, the session numbers in PTZ kindling, and the MTT test. The Student's t-test was used to analyze the behavioral hyperexcitability

test, open-field test, elevated plus maze test, and the total seizure burden in the PTZ kindling, evaluated as the area under the curve (AUC) of seizure scores across sessions. The Chi-square test was used to determine the number of animals that developed tonic-clonic behavioral seizures in the PTZ challenge. One-way ANOVA was used to analyze the sequencing data. Data were analyzed using GraphPad Prism software (version 5.0, GraphPad Software) and determined for *p < 0.05, **p < 0.01, and ***p < 0.001.

## Results

### Acute PTZ-induced seizures cause a transient increase in the Mbd3 protein level in the entorhinal cortex/amygdala

PTZ-induced seizures were used to investigate whether an acute seizure affects *Mbd3* mRNA and Mbd3 protein levels *in vivo*. The experimental design is presented in Fig 1A.

We did not detect changes in *Mbd3* mRNA in the hippocampus, entorhinal cortex/amygdala, and somatosensory cortex at 1, 4, 8, 24, and 48 h after PTZ-induced seizure ($n = 16$, Fig 1B- 1D).

The Mbd3 protein level was checked at 4, 8, 24, and 48 h after PTZ-induced seizure. Interestingly, we observed that the acute seizure caused an increase in Mbd3 protein level in the entorhinal cortex/amygdala 4 hours after the seizure. A Mann–Whitney $U$ test confirmed a statistically significant difference between the PTZ and control groups ($U = 71$, $p = 0.0318$, $n = 16$). Mean Mbd3 protein levels were higher in the PTZ group (mean ± SD: 3.6 ± 2,1) compared to the control group (2.1 ± 0.9) at this time point (Fig 1F, 1H). No statistically significant differences were observed in the entorhinal cortex/amygdala at the other time points: 8, 24, and 48 h after seizure induction (Fig 1F) or in samples from the hippocampus and somatosensory cortex at any investigated time points (Fig 1E, 1G).

### AAV-viral constructs efficiency *in vitro* and *in vivo*

Commercially prepared AAV for overexpression of Mbd3 contained MBD3 protein under the control of the synapsin promoter. The experimental design for evaluating construct efficiency *in vitro* is presented in Fig 2A. The efficiency of transducing neurons *in vitro* was 81.3 ± 9.6% for SYN-MBD3-GFP and 86.3 ± 8.3% for SYN-GFP. An increased Mdb3 protein level was observed in the cell culture transfected with AAV for Mbd3 overexpression (SYN-MBD3-GFP) compared to the control AAV (SYN-GFP) (Fig 2B). No changes in cell viability *in vitro* were observed after transfection with AAV for Mbd3 overexpression (SYN-MBD3-GFP) compared to the control AAV (SYN-GFP), checked at the same time point (Mann–Whitney U test; $U = 4$, $p > 0.9999$, $n = 3$; median 95.45 for SYN-GFP and 103.7 for SYN-MBD3-GFP)(Fig 2C).

The expression of the AAV transgene used in the study was observed *in vivo* following injection into the amygdala. Exogenous Mbd3 protein tagged with GFP reporter protein and produced by AAV SYN-MBD3-GFP or a control virus (SYN-GFP) with GFP reporter protein was observed only in neurons (Fig 3). The volume of SYN-MBD3-GFP spread was 370,8 ± 29 µm3. The product of SYN-MBD3-GFP was detected in 75.6 ± 6.1%, and SYN-GFP in 71.8 ± 9.2% of neurons. Fig 3.

### Mbd3 overexpression levels in the amygdala selectively influence animal behavior

To test the influence of Mbd3 overexpression in the amygdala on behavior, we injected an AAV for Mbd3 overexpression or control AAVs into the rat BLA. After the recovery period, the following behavioral tests were performed: the behavioral hyperexcitability test (day 14 after AAV injection), the open field test (day 17 after AAV injection), and the elevated plus maze test (day 18 after AAV injection). The experimental design is presented in Fig 4A.

Animals with Mbd3 overexpression spent less time in the outer zones of the open field arena compared to control animals (SYN-GFP: 1041 ± 20.6 s vs. SYN-MBD3-GFP: 990.5 ± 14.2 s; t-test; $t(38) = 2.033$, $p = 0.049$, $n = 20$)(Fig 4B), and correspondingly more time in the inner zone (SYN-GFP: 143.3 ± 18.2 s vs. SYN-MBD3-GFP: 192.3 ± 13.1 s; t-test; $t(38) = 2.186$, $p = 0.035$; $n = 20$)(Fig 4C).

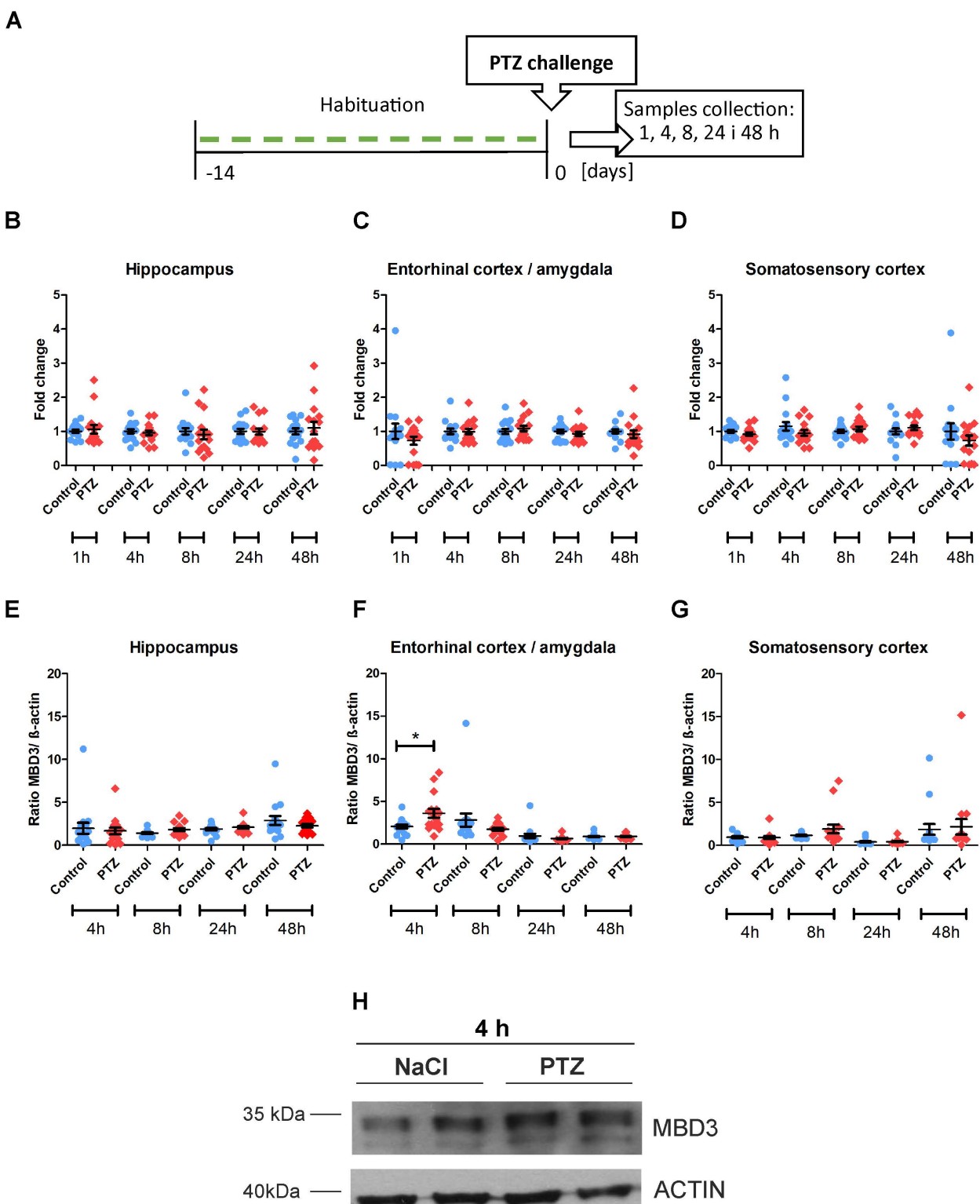

**Fig 1. Mbd3 mRNA (B-D) and protein levels (E-H) following PTZ-induced seizure. (A)** Experimental design; (B, C, **D)** *Mbd3* mRNA level in the hippocampus, the entorhinal cortex and amygdala, and somatosensory cortex, respectively; (E, F, **G)** Mbd3 protein levels in the hippocampus, the entorhinal cortex and amygdala, and the somatosensory cortex, respectively; (H) representative images of Western Blot presenting Mbd3 protein in the enthorinal cortes/amygdala at 4 hours after PTZ-induced seizure and the corresponding actin; n = 16, Mann Whitney test, *$p < 0.05$.

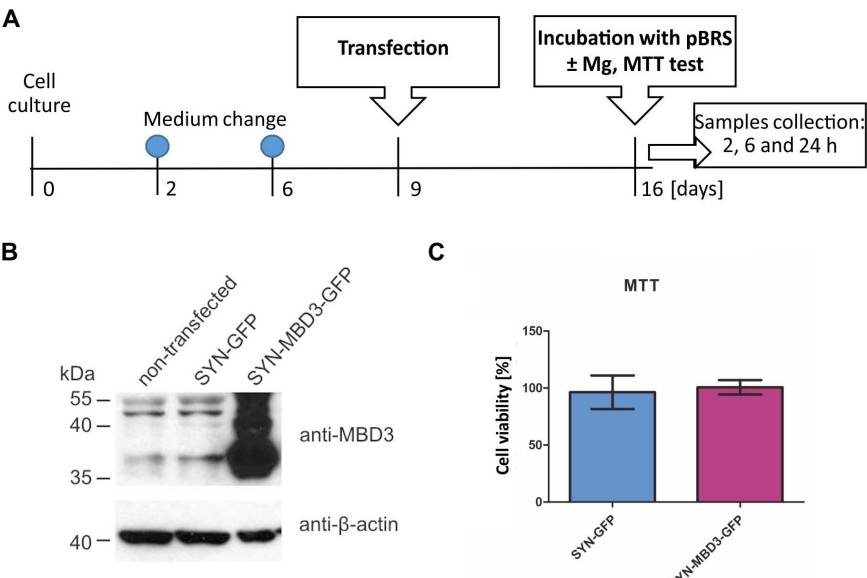

**Fig 2. The effectiveness and toxicity of MBD3 overexpressing AAV *in vitro*. (A)** Experimental design; **(B)** Mbd3 protein levels two weeks after infection with AAV for overexpression (SYN- MBD3-GFP) or the corresponding control virus (SYN-GFP); **(C)** Cell viability two weeks after transfection with AAV; MTT scores were calculated from three independent cultures (mean±SEM, Mann–Whitney U test, *p*>0.05).

No differences were observed between groups in behavioral hyperexcitability and elevated cross-maze tests (p>0.05).

## Mbd3 overexpression does not influence tonic-clonic seizures in a model of acute seizure induced by intraperitoneal injection of PTZ

We performed the PTZ challenge test to determine whether changes in the Mbd3 protein level affect seizure threshold. The experimental design is presented in Fig 4A.

80% (*n*=8) of the animals with Mbd3 overexpression developed generalized tonic-clonic seizures, while 20% (*n*=2) did not. Ninety percent of control animals (SYN-GFP, n=9) developed generalized tonic-clonic seizures, and 10% (*n*=1) did not. A chi-square test revealed no significant association between Mbd3 overexpression and seizure occurrence ($\chi^2(1)$ = 0.244, *p*=0.62). We did not observe differences in latency time to electrographic seizures between animals with Mbd3 overexpression and controls (Mann-Whitney test, *p*>0.5, *n*=10) (Fig 4D).

## Mbd3 overexpression accelerates epileptogenesis in the PTZ kindling model

To evaluate whether Mbd3 overexpression influences epileptogenesis, we performed kindling using repeated PTZ injections in rats overexpressing Mbd3 and control animals (*n*=10). The experimental design is presented in Fig 5A.

A statistically significant difference was observed in session 4, where the seizure score in the SYN_MBD3_GFP group (4.20±0.42, *n*=10) was significantly higher than in the SYN_GFP group (3.30±1.42, *n*=10), with *U*=13.5 and *p*=0.013. Similarly, in session 5, the SYN_MBD3_GFP group (4.85±0.18) showed significantly greater seizure severity than the SYN_GFP group (3.65±1.60), with *U*=8.5 and *p*=0.0026. No statistically significant differences were found in other sessions (*p*>0.05 for all) (Fig 5B). To assess total seizure burden throughout kindling, the area under the curve (AUC) was calculated for each individual. AUC values showed a significant group difference (*t*(16) = 2.65, *p*=0.019), with higher total seizure severity in the SYN_MBD3_GFP group. Animals with Mbd3 overexpression reached the criterion faster than control animals (SYN-GFP: 13.9±2.9 vs. SYN-MBD3-GFP: 6.3±3.1; *t*(16) = 2.415, *p*=0.0281) (Fig 5C). Animals with Mbd3 overexpression (SYN-MBD3-GFP)

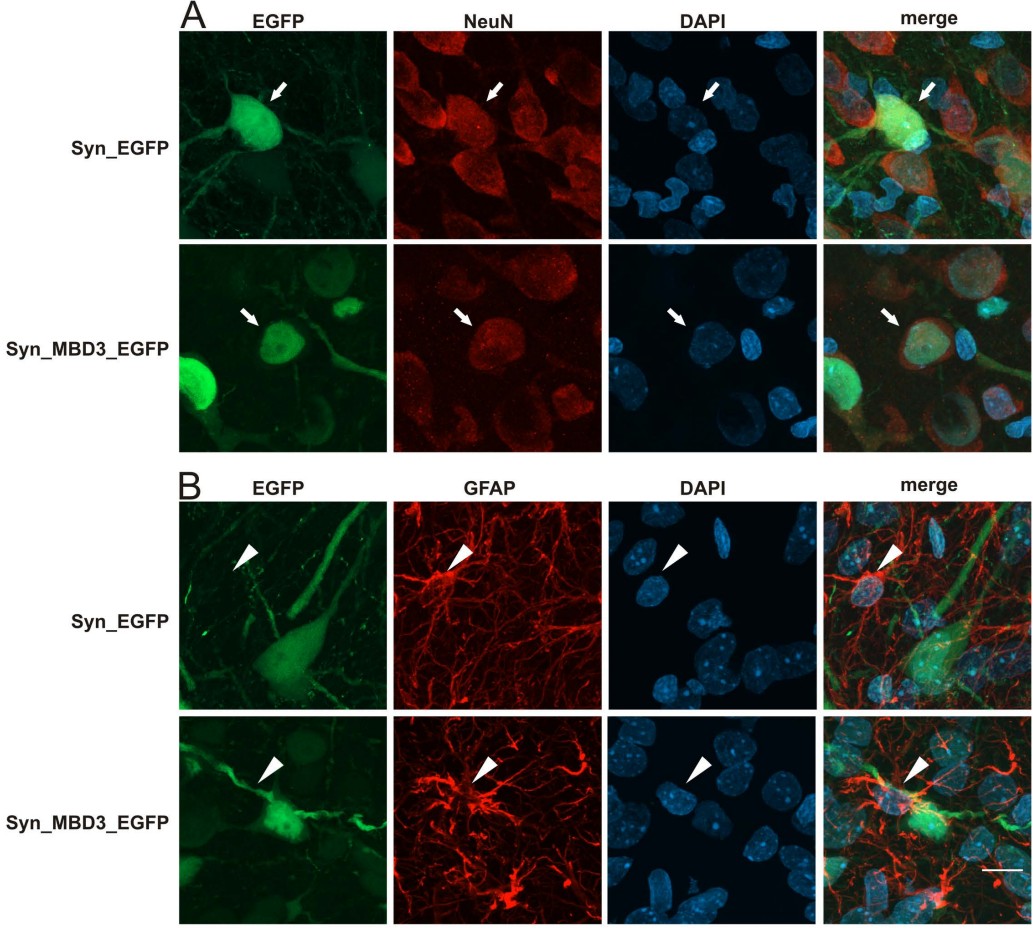

**Fig 3. Cellular localization of control AAV (SYN-eGFP) and Mbd3-overexpressing AAV (SYN-MBD3-eGFP) in the rat basolateral amygdala (BLA) *in vivo*. (A)** Representative immunofluorescence images show neuronal expression of the control AAV (SYN-eGFP, green, upper row) and the Mbd3-overexpressing AAV (SYN-MBD3-eGFP, green, lower row). Co-localization with the neuronal marker NeuN (red) confirms neuron-specific expression. Arrows indicate representative neurons co-expressing eGFP and NeuN; **(B)** Representative images of SYN-eGFP (upper row) and SYN-MBD3-eGFP (lower row) AAVs co-stained with the astrocyte marker GFAP (red). Note the absence of co-localization between eGFP (green) and GFAP (red), indicating a lack of expression in astrocytes. Arrows denote eGFP-expressing neurons; arrowheads indicate astrocytic cell bodies. All sections were counterstained with DAPI (blue); scale bar: 10 μm.

required 3.4±0.6 sessions to reach a criterion of kindling, while control animals (SYN-GFP) required 11.5±3.4 sessions. This difference did not, however, reach statistical significance (Mann-Whitney U test; $U=20.50$, $p=0.0832$ (Fig 5D). We observed that a lower number of sessions with seizure scores 4/5 to reach the criterion or death is required in MBD3-overexpressing animals compared to the control (SYN-GFP: 6.8±1.3 vs SYN-MBD3-GFP: 3.7±1.3; Mann-Whitney U test, $U=23.00$, $p=0.0385$) (Fig 5E). Moreover, there was a significant difference in the number of sessions form the first 4/5 seizure to reach the kindling criterion (SYN-GFP: 11.5±10.7 vs SYN-MBD3-GFP: 3.4±1.9; Mann-Whitney U test, $U=21.00$, $p=0.0276$).

## Epileptiform activity *in vitro* does not influence Mbd3 expression

To examine the effect of epileptiform activity on Mbd3 expression level, primary neuronal cell cultures were subjected to the medium without magnesium, inducing epileptiform discharges, and compared to physiological conditions (medium with magnesium).

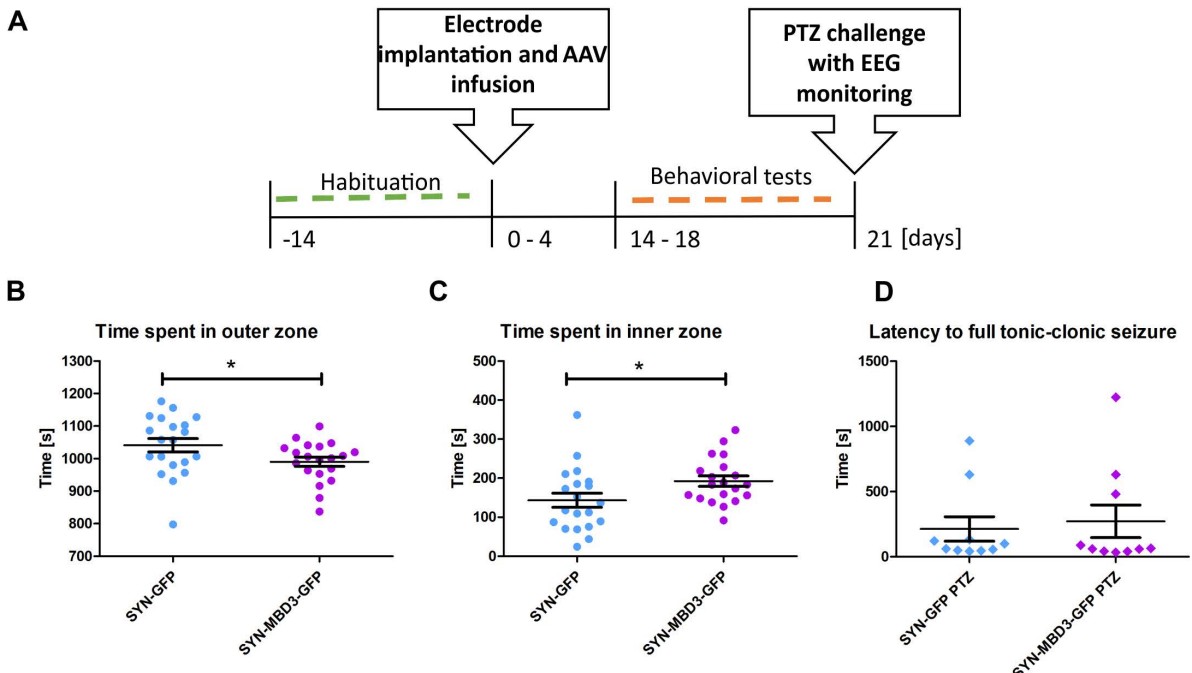

**Fig 4. Open field test performance and PTZ induced acute seizures upon Mbd3 overexpression. (A)** Experimental design; (B) time spent in the outer zone; (C) time spent in the inner zone; (D) latency time to the electrographic seizure evoked by PTZ injection (mean ± SEM, t test for B and C, Mann-Whitney $U$ test for D, $n = 20$, *$p < 0.05$).

As expected, an increase in *Mbd3* mRNA expression was observed in cells transfected with the AAV virus for Mbd3 overexpression under physiological conditions (SYN-GFP 2 h: 12.0 ± 0.1 vs. SYN-MBD3-GFP 2 h: 18.5 ± 0.1; $F_{(4,10)}$ = 254.54, $p < 0.0001$; SYN-GFP 6 h: 12.3 ± 0.1 vs. SYN-MBD3-GFP 6 h: 17.8 ± 0.1; $F_{(4,10)}$ = 434.17, $p < 0.0001$; SYN-GFP 24 h: 12.2 ± 0.0 vs. SYN-MBD3-GFP 24 h: 16.6 ± 0.5; $F_{(4,10)}$ = 45.67, $p = 0.0008$) as well as under epileptiform conditions at 2, 6, and 24 h after induction of epileptiform discharges (SYN-GFP 2 h: 11.6 ± 0.1 vs. SYN-MBD3-GFP 2 h: 18.3 ± 0.4; $F_{(4,10)}$ = 537.37, $p < 0.0001$; SYN-GFP 6 h: 12.4 ± 0.2 vs. SYN-MBD3-GFP 6 h: 17.9 ± 0.1; $F_{(4,10)}$ = 782.44, $p < 0.0001$; SYN-GFP 24 h: 12.1 ± 0.1 vs. SYN-MBD3-GFP 24 h: 16.5 ± 0.6; $F_{(4,10)}$ = 70.50, $p = 0.001$). The induction of epileptiform discharges did not affect *Mbd3* mRNA expression levels in cells transfected with AAV for Mbd3 overexpression, nor in cells transfected with control AAV (Fig 6A).

## Overexpression of Mbd3 induces changes in gene expression in a time- and state-specific manner *in vitro*

We compared gene expression changes in cultures overexpressing Mbd3 (SYN-MBD3-GFP) with those after transfection with control AAV (SYN-GFP). We checked changes 2, 6, and 24 h after induction of epileptiform discharges (induction of epileptiform discharges by medium without magnesium) or physiological conditions (medium with magnesium). A heat map presenting expression signatures is presented in Fig 6B. The functional analysis was performed with DAVID (https://david.ncifcrf.gov/) using the Gene Ontology analysis tool. In this description, we pay attention only to the most interesting or relevant findings related to neuronal function. The results of the GO terms analysis and gene names are presented in S1 Table in the Supporting Information.

At 2 hours after inducing epileptiform discharges, we observed increased expression of 39 genes and decreased expression of 1 gene in cultures overexpressing Mbd3 compared to cultures transfected with control AAV. Gene Ontology term analysis by DAVID revealed that genes increasing the expression level upon Mbd3 overexpression at two hours following epileptiform discharges represent biological functions relevant to the neuronal function, e.g., neuron projection

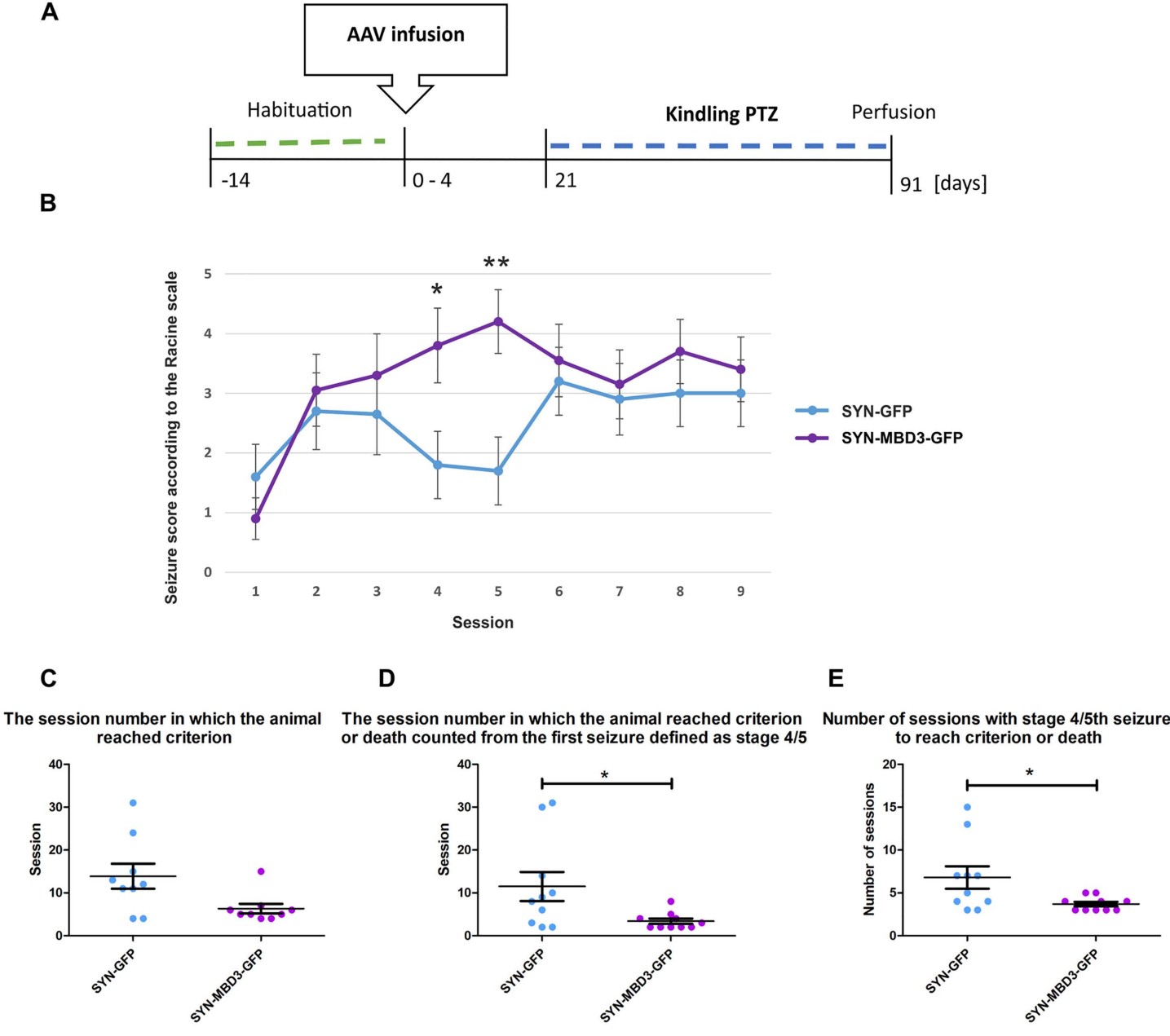

**Fig 5. Effect of Mbd3 overexpression on PTZ kindling. (A)** Experimental design; **(B)** Seizure development in control animals (SYN-GFP) and animals with Mbd3 overexpression (SYN-MBD3-GFP) during first nine kindling sessions; **(C)** The session number in which animals reached criterion; **(D)** The session number in which animal reached the criterion or death counted from the first score 4/5 seizure; (E) number of sessions with a score 4/5 until the criterion or death. Data presented as mean±SEM (Mann-Whitney $U$ test, $n = 9$ for C and $n = 10$ for B, D, **E)**, *$p < 0.05$, **$p < 0.01$).

development (LAMB2, RAB13, APOE, FGFR1), peripheral nervous system axon regeneration (MMP2, APOE), axon guidance (NOTCH3, LAMB2, PTPRV), midbrain development (MSX1, FGFR1).

In contrast, there were 142 up-regulated genes and 12 down-regulated genes under physiological conditions. Large number of upregulated genes is involved in the negative (MBD3, WWTR1, NOTCH3, ZFP217, NRARP, KLF3, FOXO1,

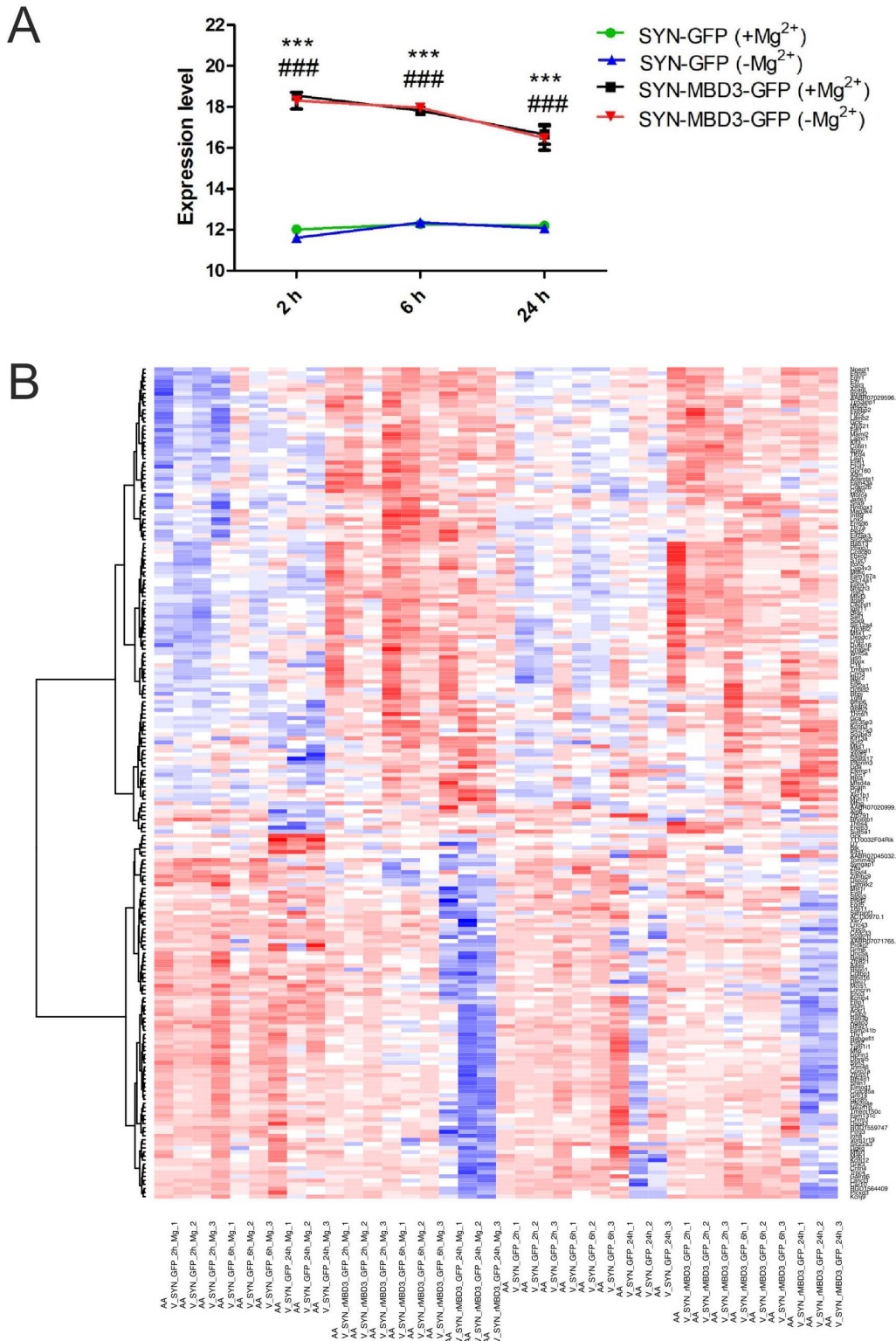

**Fig 6. The expression of *Mbd3* mRNA (A) and the Heatmap (A) of global gene expression changes after induction of epileptiform discharges in primary cortical cultures transfected with AAV for Mbd3 overexpression (SYN-MBD3-GFP) or control AAV (SYN-GFP). (A)** Mbd3 mRNA expression 2, 6, and 24 hours after induction of epileptiform discharges. Cells overexpressing Mbd3 (SYN-MBD3-GFP) under physiological conditions

(*** $p < 0.001$) and after induction of epileptic-like discharges (### $p < 0.001$) compared to samples transfected with control virus (SYN-GFP). (-Mg2+) – conditions inducing epileptiform discharges; (+Mg2+) – physiological conditions; (mean ± SEM, one-way ANOVA). **(B)** The heatmap of gene expression changes detected at 2, 6, and 24 hours after induction of epileptiform discharges in cells overexpressing Mbd3 vs. controls. Each row in the heatmap represents a single gene, and each column represents a corresponding sample. A warm (red) color indicates a higher expression level, while a cold (blue) color indicates a lower gene expression level. Individual clusters in panel B contain groups of genes with similar expression profiles. The red color indicates genes with the highest value, while shades close to red indicate a high value. In contrast, shades of blue and green indicate the lowest value of membership in a given cluster. (-Mg$^{2+}$) – epileptiform discharges; (+Mg$^{2+}$) – physiological conditions, h – hours, SYN-GFP – control AAV, SYN-MBD3-GFP - AAV for Mbd3 overexpression;.

NFKB1, SIRT2, HOPX, HIPK2, EDNRB, SALL1, SMO, ID1, SOX9, MSX1, EZR, MXD1, FGFR2, FGFR1) or positive (WWTR1, NOTCH3, CHD7, RORB, LITAF, ETV4, FOXO1, NFKB1, SIRT2, HIPK2, SALL1, SMO, RFX4, S1PR1, REL, SOX9, MSX1, BMPR1B, PLPP3, IER5, FGFR2, NFE2L2, FGFR1) egulation of transcription by RNA polymerase II, negative regulation of apoptotic process (NQO1, CDKN1A, SMAD6, FOXO1, NFKB1, SIRT2, GCLC, EDNRB, SMO, ID1, SPRY2, SOX9, MSX1, BMPR1B), brain-derived neurotrophic factor receptor signaling pathway (DDR1, FGFR2, EPHA2, FGFR1), neuron projection regeneration (SMO, ADM). Down-regulated genes represented, e.g., regulation of transcription by RNA polymerase II (USP22, MED4) and establishment of cell polarity (SLC9A6, RHOBTB1). The expression of 25 genes was increased by Mbd3 overexpression in both epileptiform and physiological conditions, indicating they are strongly dependent on Mbd3. These genes represented following relevant Biological Function Gene Ontology terms: axon guidance (NOTCH3, LAMB2, PTPRV), neuron projection development (LAMB2, RAB13, FGFR1), negative regulation of transcription by RNA polymerase II (MBD3, NOTCH3, MSX1, SIRT2, FGFR1), midbrain development (midbrain development), cell adhesion (LAMB2, PBXIP1, HAPLN3).

Six hours after induction of epileptiform discharges, we observed one up-regulated gene (SCUBE3) and one down-regulated gene (NF2) in cultures overexpressing Mbd3 compared to control AAV. Under physiological conditions, 165 genes were up-regulated and 57 were down-regulated. Upregulated genes represented several relevant functions including axon guidance (EFNB1, NOTCH3, EDNRA, NRP2, UNC5B, WNT5A, LAMC1, PTPRV, NECTIN1), positive regulation of gene expression (WNT5A, EIF2AK3, NFATC1, RBPJ, PKD2, DNAJA4, SH3PXD2B, AGO2, TP53INP1, SOX9, FGFR2, NFE2L2, PPARD), negative regulation of transcription by RNA polymerase II (NOTCH3, ZHX2, GLIS3, NFATC1, RBPJ, KLF3, HOPX, AMOT, HMBOX1, SOX9, MSX1, FGFR2, FGFR1, PPARD), regulation of transcription by RNA polymerase II (ZHX2, ZFP521, MIDEAS, TSHZ1, JADE1, GLIS3, NFATC1, RORA, RORB, RBPJ, KLF3, CDC14A, HOPX, MED13L, SOX9, MSX1, ZFP652, NFE2L2, PPARD). Downregulate genes represented synaptic vesicle endocytosis (DENND1A, AP2A1, AP2A2), synaptic vesicle lumen acidification (ATP6AP1, ATP6V0A1) or neuron projection development (CYFIP2, NCDN, PPP1R9B). Interestingly, both genes altered by Mbd3 overexpression in epileptiform conditions (SCUBE3 and NF2) were affected in the same direction in physiological conditions, indicating a crucial role of Mbd3 in the regulation of their expression.

A24 hours after inducing epileptiform discharges, we identified seven upregulated genes and ten downregulated genes in cultures overexpressing Mbd3 compared to control AAV. Upregulated genes are involved in the negative regulation of the neuron apoptotic process (FYN, CRLF1) or cell differentiation (FYN, HOPX), while downregulated genes are involved in the action potential (KCNMB2), potassium ion transmembrane transport (KCNMB2), and other processes. Under physiological conditions, 240 genes were up-regulated and 485 genes were down-regulated. Functions of upregulated genes included cell adhesion (FARP2, APP, BCAM, CDH2, CLDN14, SDC1, PRKCA, ITGAV, HAPLN3, THBS4, VCL, FREM3), forebrain development (APP, NOTCH3, IFT88, ZEB1, CNP, OGT), glial cell differentiation (C1S, CDH2, DLG5, HAPLN3), negative regulation of neuron differentiation (APP, NOTCH3, OLIG2, GPR37L1, RTN4), neuron remodeling (FARP2, APP, C1QL1), nervous system development (EFNB1, IFT88, GFRAL, SLC1A2, KREMEN1, OLIG2, THBS4, RTN4), neuron projection development (APP, GPRIN2, SNX1, PTPRZ1, PTPRM, FGFR1), regulation of synapse organization (WNT5A, ETV5, NECTIN1), or positive regulation of gene expression (APP, GSN, CSF1, HFE, WNT5A, PKD2, BMP6, VSIR, TGFBR3, NWD1,

GAS6, OGT, TLR3). The top functions represented by genes downregulated in physiological conditions were related to regulation of postsynapse assembly (CYFIP2, PSD, ARHGAP33, PALM, NEURL1, DCX, SYNDIG1, RAC3, CRMP1, ASIC2, CC2D1A), potassium ion transmembrane transport (KCNJ4, KCNH2, HCN3, KCNH3, KCNIP1, KCNIP2, KCNJ9, KCNJ12, KCNIP3, KCNIP4, KCNC4, SLC9A4, KCNT2, EEF1A2, SLC17A7, KCNK4), neuron projection development (CYFIP2, PSD, UNC5A, STMN2, STMN3, STMN4, CAMSAP3, SAMD14, ATCAY, GPRIN1, STMN1, RAC3, MAPT, CNTN4, NEFH, CDK5R1), axonogenesis (SEMA4A, SHTN1, UCHL1, SYNGAP1, RAB3A, STMN1, APLP1, TRIM46, CCK, ADCY1, PIP5K1C, MAPT), regulation of postsynaptic membrane potential (GRIN3A, FGF14, CHRM1, GABRA3, KCNC4, GRIK3, GRIN2B, GRM1), regulation of neuron migration (CAMK2B, SHTN1, CAMK2A, RAC3, NSMF, SCRT1), chemical synaptic transmission (PRKCG, CACNB3, GRIN3A, FGF14, CHRM1, GRM6, GABRA3, GRIK3, SLC17A7, SNCB, GRIN2B, GRM1), and axon guidance (CYFIP2, SEMA4A, UNC5A, SEMA4F, EFNA2, TUBB3, RTN4RL1, RAC3, EVL, CNTN4, DSCAML1, CDK5R2, CDK5R1). Five genes showed increased expression (AKR1B1, MYH11, HOPX, CRLF1, SCUBE3), and three genes showed decreased expression (LOCRIN, NF2, TCP11) under both conditions.

Interestingly, two genes, SCUBE3 and NF2, were regulated by Mbd3 at 6 and 24 h time points, irrespective of physiological vs. epileptiform conditions.

In this manuscript, we focus on presenting ensembles of genes whose expression changes similarly. Unsupervised clustering analysis was performed to check that. Genes were clustered into 9 clusters, each grouping genes with similar patterns of changes in expression levels depending on conditions: physiological (medium with magnesium) vs. after induction of epileptiform discharges (medium without magnesium), Mbd3 expression level (overexpression of Mbd3 vs. control AAV), and the time since the induction of epileptiform discharges (2, 6, and 24 h). The clusters resulting from the analysis are presented in Fig 7.

We observed that clusters 4, 6, 7, 8, and 9 share some similarities in gene expression patterns. These clusters group genes whose expression differs between cultures overexpressing Mbd3 and cultures transfected with control AAV, regardless of the conditions. An increase in expression level was observed in clusters 4, 7, and 9 due to Mbd3 overexpression. Interestingly, the Mbd3 gene was assigned to cluster 7. In contrast, a decrease in expression level was observed in clusters 6 and 8. The effect of epileptiform discharges was less noticeable. GO (Gene Ontology) term analysis of the genes assigned to each cluster carried out in the DAVID database (https://david.ncifcrf.gov/) functions Wnt signaling (Cluster 4, 6, 7), Notch signaling (Cluster 4), gene expression regulation (Cluster 4, 6, 7, 8, 9), dendritic and axonal plasticity (4, 6, 7, 9), development and differentiation (4, 7, 9). apoptosis (4, 7, 8), cell motility and adhesion (Cluster 7), and other (Table 1 and S2 Table in Supporting Information).

In clusters 2 and 3, epileptiform discharges induced a similar pattern to that observed with Mbd3 overexpression under physiological conditions. It suggests that, in this instance, Mbd3 mimicked the effects of epileptic discharges. According to the DAVID analysis, GO functions represented by genes in Cluster 1 were related to regulation of synaptic plasticity, axogenesis Genes in Cluster 2 were assigned to several GO terms including G protein-coupled glutamate receptor signaling pathway, potassium ion import across plasma membrane, calcium ion transmembrane transport and axon guidance, that are all crucial for proper neuronal functioning (Table 1 and S2 Table in Supporting Information).

Cluster 5 included genes with downregulated expression levels in cultures overexpressing Mbd3 under physiological conditions. Induction of epileptiform discharges prevented this downregulation. Functional annotation indicated a representation of genes related to transcription regulation, neuronal development, and BDNF signaling (Table 1, cluster 5).

## Discussion

In the present study, we demonstrated for the first time that (I) seizures cause a transient, brain area-specific increase in Mbd3 protein levels *in vivo*; (II) modification of Mbd3 expression in a rat's amygdala induces abnormal anxiety responses;

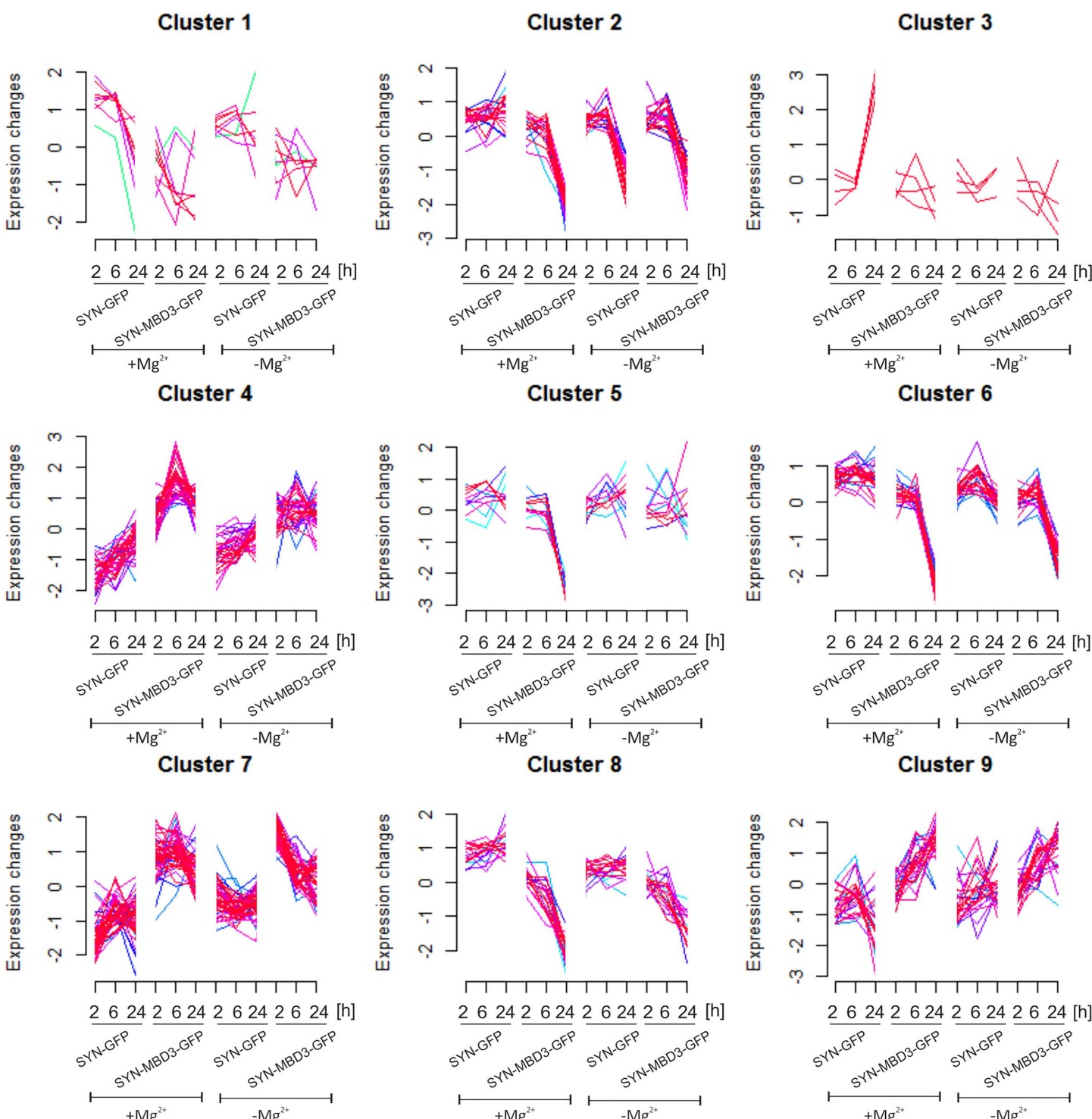

**Fig 7. Cluster analysis of gene expression changes at different time points after induction of epileptiform discharges in primary cortical cultures transfected with AAV for Mbd3 overexpression (SYN-MBD3-GFP) or control AAV (SYN-GFP).** The basic expression level of the investigated genes was calculated from non-transfected cell cultures incubated under physiological conditions ($+Mg2+$) or with increased epileptic-like activity ($-Mg^{2+}$). Individual clusters contain groups of genes whose expression profiles are similar. The red color indicates genes with the highest value, while shades close to red indicate a high value. In contrast, shades of blue and green indicate the lowest value of membership in a given cluster. ($-Mg^{2+}$) – epileptiform discharges; ($+Mg^{2+}$) – physiological conditions, h – hours, SYN-GFP – control AAV, SYN-MBD3-GFP - AAV for Mbd3 overexpression.

**Table 1. Gene Ontology Biological Process Terms represented by genes in each Cluster.**

| Gene ontology term | Official gene name |
|---|---|
| **Cluster 1** | |
| regulation of long-term neuronal synaptic plasticity | SYNGAP1 |
| regulation of synapse structure or activity | SYNGAP1 |
| regulation of synaptic plasticity | SYNGAP1 |
| modulation of chemical synaptic transmission | SYNGAP1 |
| axonogenesis | SYNGAP1 |
| negative regulation of the neuron apoptotic process | SYNGAP1 |
| CAMKK-AMPK signaling cascade | CAMKK2 |
| regulation of transcription by RNA polymerase II | USP22 |
| **Cluster 2** | |
| **adenylate cyclase-inhibiting G protein-coupled glutamate receptor signaling pathway** | **GRM6, GRIK3** |
| **G protein-coupled glutamate receptor signaling pathway** | **GRM6, GRIK3** |
| **regulation of monoatomic ion transmembrane transport** | **KCNJ9, KCNJ12** |
| **potassium ion import across plasma membrane** | **KCNJ9, KCNJ12** |
| **regulation of presynaptic membrane potential** | **KCNJ9, GRIK3** |
| **regulation of cytosolic calcium ion concentration** | **TRPC5, TRPC4** |
| calcium ion transmembrane transport | TRPC5, TRPC4 |
| potassium ion transmembrane transport | KCNJ9, KCNJ12 |
| chemical synaptic transmission | GRM6, GRIK3 |
| axon guidance | UNC5A, CNTN4 |
| neuron projection development | UNC5A, CNTN4 |
| apoptotic process | UNC5A, MTFP1 |
| G protein-coupled receptor signaling pathway | PROKR2, GPR85 |
| **Cluster 4** | |
| positive regulation of gene expression | DNAJA4, WNT5A, TP53INP1, EIF2AK3, SOX9, RBPJ, EZR |
| **stem cell proliferation** | **WNT5A, SOX9, RBPJ** |
| **olfactory bulb interneuron development** | **SALL3, WNT5A** |
| **canonical Wnt signaling pathway** | **EDNRB, WNT5A, SOX9** |
| **phospholipase C-activating G protein-coupled receptor signaling pathway** | **EDNRB, WNT5A, NTSR2** |
| **negative regulation of apoptotic process** | **EDNRB, WNT5A, EIF2AK3, TGFA, SOX9** |
| **cellular response to transforming growth factor beta stimulus** | **WNT5A, SOX9, ZFP36L2** |
| **negative regulation of transcription by RNA polymerase II** | **HMBOX1, EDNRB, SOX9, RBPJ, EZR, HOPX** |
| **Notch signaling pathway** | **MAML2, SOX9, RBPJ** |
| **positive regulation of DNA-templated transcription** | **HMBOX1, WNT5A, TP53INP1, SOX9, RBPJ** |
| **positive regulation of protein phosphorylation** | **EDNRB, WNT5A, SOX9** |
| **cell differentiation** | **WNT5A, SOX9, RBPJ, HOPX** |
| **positive regulation of dendrite extension** | **SLC23A2, WNT5A** |
| neuron differentiation | WNT5A, RBPJ, LNX2 |
| negative regulation of DNA-templated transcription | HMBOX1, WNT5A, SOX9, RBPJ |

*(Continued)*

**Table 1.** (Continued)

| Gene ontology term | Official gene name |
|---|---|
| **Cluster 5** | |
| positive regulation of neuron projection development | SERPINF1 |
| short-term memory | SERPINF1 |
| nervous system development | MST1R |
| brain-derived neurotrophic factor receptor signaling pathway | MST1R |
| positive regulation of neurogenesis | SERPINF1 |
| regulation of transcription by RNA polymerase II | RGD1559747 |
| **Cluster 6** | |
| **microtubule bundle formation** | **TRIM46, AAAS** |
| axonogenesis | SHTN1, TRIM46 |
| cellular response to insulin stimulus | H2AZ1, GRB14 |
| Wnt signaling pathway | TGFB1I1, RSPO1 |
| positive regulation of transcription by RNA polymerase II | H2AZ1, MLIP |
| **Cluster 7** | |
| cell migration | ERRFI1, LAMB2, S1PR1, FAT1, PBXIP1, ITGAV |
| **midbrain development** | **MSX1, FGFR2, FGFR1** |
| **fibroblast growth factor receptor signaling pathway involved in orbitofrontal cortex development** | **FGFR2, FGFR1** |
| **negative regulation of extrinsic apoptotic signaling pathway** | **GCLC, ITGA6, ITGAV** |
| **positive regulation of Wnt signaling pathway** | **SALL1, PBXIP1, FGFR2** |
| **ventricular zone neuroblast division** | **FGFR2, FGFR1** |
| **positive regulation of transcription by RNA polymerase II** | **NOTCH3, SALL1, CHD7, S1PR1, ITGA6, MSX1, FGFR2, NFE2L2, FGFR1** |
| **olfactory nerve development** | **SALL1, CHD7** |
| **positive regulation of neuron projection development** | **ADAMTS1, ITGA6, NFE2L2, FGFR1** |
| **orbitofrontal cortex development** | **FGFR2, FGFR1** |
| **cell adhesion** | **LAMB2, PBXIP1, ITGAV, THBS4, FREM3** |
| **negative regulation of transcription by RNA polymerase II** | **MBD3, NOTCH3, SALL1, MSX1, KLF3, FGFR2, FGFR1** |
| negative regulation of DNA-templated transcription | GCLC, SALL1, MDFIC, PBXIP1, MSX1 |
| neuron projection development | LAMB2, RAB13, FGFR1 |
| positive regulation of DNA-templated transcription | SALL1, MDFIC, JADE1, PBXIP1, NFE2L2 |
| neuron fate commitment | NOTCH3, ZFP521 |
| **Cluster 8** | |
| cytoskeleton organization | LORICRIN, THY1 |
| negative regulation of apoptotic process | THY1, PIM2 |
| positive regulation of DNA-templated transcription | MCRS1, PIM2 |
| **Cluster 9** | |
| negative regulation of neuron projection development | EFEMP1, RTN4 |
| response to activity | RTN4, BMP6 |
| transmembrane transport | MFSD4A, SLC35E3 |

*(Continued)*

**Table 1.** (Continued)

| Gene ontology term | Official gene name |
|---|---|
| neuron differentiation | RTN4, BMP6 |
| positive regulation of transcription by RNA polymerase II | IKZF2, PKD2, BMP6 |
| positive regulation of cell population proliferation | EFEMP1, CRLF1 |
| positive regulation of gene expression | PKD2, BMP6 |

Functional annotation was performed using the Functional Annotation Chart tool in DAVID (https://david.ncifcrf.gov/). GO (Gene Ontology) terms enriched in the given cluster (p<0.05) are highlighted in bold. Up to 15 selected GO terms relevant to neuronal functions are presented. The list of genes with full gene names and respective complete lists of GO terms is presented in S2 Table in the Supporting Information.

(III) overexpression of Mbd3 *in vivo* accelerates the development of kindling; (IV) Mbd3 is involved in the regulation of gene expression during *in vitro* Epileptiform discharges occur in a manner consistent with this protein's potentially proepileptic role.

We investigated the effects of increased Mbd3 levels in the brain on the animal's behavior. The behavioral disturbances were minor and highly specific, evident only in the open field test. Rats with Mbd3 overexpression spent less time in the outer zone and more time in the arena's inner zone compared to the control group, indicating reduced anxiety [44]. The location of the AAV injection may explain the specificity of the observed behavioral dysfunction since AAV constructs were injected into the amygdala, a structure essential for anxiety regulation [45,46]. Our results suggest that the upregulation of Mbd3 expression alters anxiety responses, possibly through changes in gene expression.

To our knowledge, there is no data on the effect of Mbd3 on anxiety behavior. However, another protein from the MBD family, MeCP2, is involved in the response to early-life stress. A study by Murgatroyd et al. [47]indicated that early-life stress in mice leads to hypomethylation of MeCP2 binding sites. It has been suggested that MeCP2 phosphorylation induces overexpression of the AVP (arginine vasopressin) gene in neurons, which is accompanied by increased activity of the hypothalamic-pituitary-adrenal cortex axis and elevated stress levels observed in the forced swimming test [47]. It is unknown whether a similar mechanism regulates anxiety behavior through Mbd3.

The Mbd3 protein has been rarely investigated in the context of epilepsy and epileptogenesis. Previously, Bednarczyk et al. showed elevated Mbd3 protein levels in the rat amygdala and cortex 14 days after initiating epileptogenesis using the amygdala stimulation model of epilepsy [23]. To determine whether the increase in Mbd3 levels was due to mechanisms involved in the process of epileptogenesis or whether spontaneous seizures triggered it, we examined whether acute seizures affected Mbd3 protein expression levels. In the PTZ challenge, we observed a transient rise in Mbd3 protein levels in the animals' entorhinal cortex and amygdala shortly after an acute seizure. A similar effect was reported by Francis et al., who identified an increase in Mbd3 mRNA levels in the hippocampus 24 hours after a tonic-clonic seizure induced by electrical stimulation of the amygdala and in perforant-pathway-kindled animals [48]. This change in expression was transient; after 28 days, Mbd3 expression levels returned to baseline [48]. In conclusion, the increase in Mbd3 levels in epileptic animals can be at least partially attributed to the occurrence of seizures.

The results of Bednarczyk et al. prompted us to investigate whether the elevated level of Mbd3 observed in an experimental model of epilepsy facilitates seizure generation [23]. Accordingly, we examined the effect of increased Mbd3 levels on seizure threshold *in vivo* using a PTZ-induced acute seizure model. We found no impact of Mbd3 overexpression on the PTZ-induced seizures. Interestingly, Wang et al. observed a seizure-suppressing effect resulting from the decrease of Mbd3 protein [25]. They demonstrated that reducing Mbd3 expression in mice with shRNA leads to a reduced incidence of generalized seizures following pilocarpine administration [25].

In the present study, we demonstrated for the first time that the overexpression of Mbd3 accelerates PTZ kindling, indicating a proepileptogenic effect. The mechanism behind Mbd3's proepileptogenic effect remains unknown. Bednarczyk et

al. observed that Mbd3 binding to DNA changes 14 days after status epilepticus [23]. For specific genes, Mbd3 binding to DNA alterations occurred in animals with spontaneous seizures [23]. This suggests that Mbd3's involvement in the epigenetic regulation of gene expression may contribute to epileptogenesis. Our results are particularly exciting in the context of Bednarczyk et al.'s study, as they may indicate that an increased level of Mbd3 exerts a proepileptogenic effect and contributes to epilepsy development.

Finally, we examined gene expression in an *in vitro* model of epileptiform discharge to determine the impact of Mbd3 protein overexpression on global gene expression. The present findings demonstrate that MBD3 overexpression induces distinct, time- and state-dependent changes in gene expression profiles under both physiological and epileptiform conditions. The observed differential gene expression patterns suggest that MBD3 functions as a molecular switch that modulates neuronal excitability through coordinated regulation of multiple gene networks.

The temporal analysis of gene expression changes following Mbd3 overexpression revealed distinct patterns at 2, 6, and 24 hours after induction of epileptiform activity. At 2 hours post-induction, cultures overexpressing Mbd3 showed a relatively modest response. The upregulated genes at this early timepoint were primarily associated with critical neuronal functions, including neuron projection development, peripheral nervous system axon regeneration, axon guidance, and midbrain development [49,50]. This early response suggests that Mbd3 overexpression rapidly activates developmental and regenerative pathways in neurons exposed to epileptiform conditions. In contrast, under physiological conditions at 2 hours, a much more robust transcriptional response was observed with 142 upregulated and 12 downregulated genes. This dramatic difference suggests that Mbd3's transcriptional regulatory effects are more pronounced under normal conditions, potentially indicating a homeostatic role in maintaining neuronal gene expression programs [51,52]. The large number of upregulated genes under physiological conditions involved both negative and positive regulation of transcription by RNA polymerase II, suggesting that Mbd3 acts as a master regulator of transcriptional networks in resting neurons. At 6 hours following epileptiform discharge induction, the transcriptional response was markedly reduced to only two differentially expressed genes. This temporal shift suggests that the initial burst of transcriptional activity at 2 hours is followed by a more selective and refined gene expression response. Under physiological conditions at 6 hours, however, the response remained robust, indicating sustained transcriptional activity in the absence of epileptiform stress. At 24 hours post-induction, the temporal pattern observed at earlier timepoints was maintained, with significantly more robust transcriptional changes occurring under physiological conditions compared to epileptiform conditions. This consistent pattern across all three timepoints suggests that epileptiform activity fundamentally constrains Mbd3's transcriptional regulatory capacity, possibly through competing chromatin remodeling mechanisms or altered cellular energy states that limit extensive gene expression reprogramming [53,54].

The persistence of SCUBE3 upregulation and NF2 downregulation across multiple timepoints reinforces them as core, time-independent Mbd3 targets. SCUBE3 (Signal peptide, CUB domain, EGF-like 3) is a secreted glycoprotein that has been implicated in various developmental processes and cell signaling pathways, and has recently been associated with schizophrenia [55,56]. The consistent upregulation of SCUBE3 may represent a protective or adaptive response to maintain neuronal network integrity under pathological conditions. NF2 (Neurofibromin 2/Merlin), conversely, showed consistent downregulation. NF2 is a tumor suppressor protein that regulates cell proliferation and contact inhibition, and its loss is associated with neurofibromatosis type 2 [57,58]. The downregulation of NF2 following Mbd3 overexpression during epileptiform activity may indicate a shift toward increased cellular proliferation or altered cell-cell contact signaling. This finding is particularly relevant given NF2's role in regulating the Hippo pathway and its connections to neuronal development and maintenance [59].

To detect common patterns of Mbd3 overexpression on gene expression, we performed unsupervised cluster analysis (Fig 7, Table 1). Clusters 4, 6, 7, 8, and 9 group genes in which changes in expression are induced by Mbd3 overexpression, and the influence of the environment (physiological vs. epileptiform discharge) is less prominent. Functional analysis revealed that these genes are involved, among others, in the Notch signaling pathway (Fig 7,

Table 1). The Notch signaling pathway regulates the development of the central nervous system [60], and it may also be involved in epilepsy. Activation of the Notch signaling pathway has been demonstrated in an experimental model of temporal lobe epilepsy induced by kainic acid injection and in tissue from patients with epilepsy [61]. Notch activation is positively correlated with increased seizure frequency, suggesting that activation of Notch signaling during epileptogenesis may increase the risk of developing epilepsy and promote the occurrence of seizures [61]. Another pathway represented in clusters 4, 6, and 7 is the Wnt signaling pathway, which is highly significant in the context of epilepsy research. Wnt signaling plays crucial roles in neuronal development, synaptic plasticity, and adult neurogenesis, and its dysregulation has been implicated in epileptogenesis [62,63]. The canonical Wnt pathway, represented by genes such as WNT5A and associated transcription factors like SOX9, regulates neuronal fate specification and dendritic arborization. Aberrant Wnt signaling can lead to altered neuronal connectivity and increased seizure susceptibility by disrupting the balance between excitatory and inhibitory neurotransmission [28,64]. The regulation of Wnt pathway genes by Mbd3 suggests that chromatin remodeling may influence seizure susceptibility through modulation of this critical developmental pathway.

Other attractive clusters are clusters 2 and 3 (Fig 7B, Table 1), in which the expression patterns of genes are altered due to Mbd3 overexpression relative to the control in the physiological environment. After the induction of epileptiform discharges, expression of genes from this cluster did not change and, regardless of the level of Mbd3, was similar to the expression pattern in physiological conditions after Mbd3 overexpression. It can be suggested that, in this case, the increased Mbd3 level mimicked the effects of epileptic discharges. Genes in these clusters are involved, among others, in potassium transport regulation (KCNJ9 and KCNJ12) and glutamate signaling. KCNJ9 and KCNJ12 encode G protein-coupled inwardly rectifying potassium (GIRK) channels that are crucial for maintaining neuronal excitability and have been directly implicated in epilepsy pathogenesis [65,66]. KCNJ9 (GIRK3) and KCNJ12 (GIRK2) form heteromeric channels that respond to neurotransmitters such as GABA and adenosine, providing inhibitory control over neuronal firing. The downregulation of these channels by Mbd3 under physiological conditions suggests a potential pro-excitatory shift in neuronal networks, which could lower seizure threshold and increase epileptic susceptibility [67]. This finding is particularly relevant given that GIRK channel dysfunction has been associated with various forms of epilepsy, including temporal lobe epilepsy and absence seizures [68,69]. GRM6 and GRIK3, encoding metabotropic glutamate receptor 6 and kainate receptor subunit 3, respectively, represent additional critical components of glutamatergic neurotransmission whose dysregulation is central to epilepsy pathophysiology [70,71]. GRM6 typically functions as an inhibitory metabotropic glutamate receptor, particularly important in retinal signaling but also present in brain regions where it modulates synaptic transmission. GRIK3 is a kainate receptor subunit that contributes to both synaptic and extrasynaptic glutamate signaling, with mutations and expression changes linked to seizure susceptibility and epileptic encephalopathies [72,73]. The coordinated downregulation of these glutamate receptors alongside potassium channels suggests that Mbd3 orchestrates a comprehensive remodeling of neuronal excitability networks.

The last cluster we highlight is cluster 5 (Fig 7, Table 1). This cluster comprises genes for which epileptiform discharges counteract the expression changes induced by Mbd3 overexpression. These are genes involved in, for example, the biosynthesis of peptide hormones. Little is known about the involvement of peptide hormones in epilepsy, but it has been shown that oxytocin inhibits PTZ-induced seizures in rats [74]. At the same time, insulin delays the development of seizures and reduces their intensity in mice [75].

A comprehensive analysis of transcriptomic data on the effects of Mbd3 levels and neuronal excitability status indicates a complex relationship. We conclude that many processes affected by Mbd3 can be potentially involved in developing epilepsy. Considering that the increase in Mbd3 protein level is proepileptic and seizures transiently increase Mbd3 protein level, we cannot exclude the creation of a vicious circle leading to the aggravation of the disease. Counteracting the proepileptic effects of increased Mbd3 levels in the brain can be proposed as a target for therapy.

## Limitations of the study

Most studies investigating experimental epileptogenesis and epilepsy use male animals [23,28,48,76]. Males and females differ in behavioral and pharmacological responses. Our study also conducted experiments using only male rats to avoid additional factors such as gender-based variability. However, it is essential to note that, regardless of the animals used in experiments, animal models provide a unique opportunity to study the independent effects of epilepsy without confounding factors [77].

Due to the limitations of the isolation technique, the Mbd3 mRNA and Mbd3 protein levels from the entorhinal cortex and amygdala were analyzed together.

This study identified that seizures might increase Mbd3 protein levels in the epileptic brain. We also demonstrate that the Mbd3 protein has pro-epileptic effects. Another interesting finding in the present study was that the Mbd3 protein regulates gene expression in a time—and state-specific manner in neurons. Moreover, we observed that the Mbd3 protein regulates the expression of many genes involved in various mechanisms potentially related to the development of epilepsy. Our results support the view that the Mbd3 protein is involved in epilepsy development.

## Supporting information

**S1 Table. The results of the GO terms analysis and gene names for the heat map presented in Fig 6B.** (ODS)

**S2 Table. The results of the GO terms analysis and gene names for the cluster analysis presented in Fig 7.** (ODS)

## Acknowledgments

The authors thank Prof. Monika Liguz-Lęcznar for constructive criticism of the manuscript and Karolina Godlewska, M.Sc., for help in figure preparation.

## Author contributions

**Conceptualization:** Katarzyna Lukasiuk.

**Data curation:** Karolina Nizinska.

**Formal analysis:** Karolina Nizinska, Sandra Binias, Dorota Nowicka, Kinga Szydlowska.

**Funding acquisition:** Katarzyna Lukasiuk.

**Investigation:** Karolina Nizinska, Maciej Olszewski, Dorota Nowicka, Kinga Szydlowska, Kinga Nazaruk, Katarzyna Lukasiuk.

**Methodology:** Maciej Olszewski, Kinga Szydlowska.

**Supervision:** Bartosz Wojtas, Katarzyna Lukasiuk.

**Writing – original draft:** Karolina Nizinska, Dorota Nowicka, Kinga Szydlowska, Bartosz Wojtas, Katarzyna Lukasiuk.

**Writing – review & editing:** Karolina Nizinska, Dorota Nowicka, Katarzyna Lukasiuk.

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
