## [Decision Letter · Decision Letter 0]

27 Jun 2025

Dear Dr. Lukasiuk,

We look forward to receiving your revised manuscript.

Kind regards,

Giuseppe Biagini, MD

Academic Editor

PLOS ONE

**Journal Requirements:**

1. When submitting your revision, we need you to address these additional requirements. Please ensure that your manuscript meets PLOS ONE's style requirements, including those for file naming. The PLOS ONE style templates can be found at https://journals.plos.org/plosone/s/file?id=wjVg/PLOSOne_formatting_sample_main_body.pdf and https://journals.plos.org/plosone/s/file?id=ba62/PLOSOne_formatting_sample_title_authors_affiliations.pdf 2. Thank you for stating in your Funding Statement: This work was supported by an OPUS grant 2015/19/B/NZ4/01401 (KL).   Please provide an amended statement that declares *all* the funding or sources of support (whether external or internal to your organization) received during this study, as detailed online in our guide for authors at http://journals.plos.org/plosone/s/submit-now.  Please also include the statement “There was no additional external funding received for this study.” in your updated Funding Statement. Please include your amended Funding Statement within your cover letter. We will change the online submission form on your behalf.

Reviewers' comments:

Reviewer's Responses to Questions

**Comments to the Author**

1. Is the manuscript technically sound, and do the data support the conclusions?

Reviewer #1: Yes

Reviewer #2: Yes

2. Has the statistical analysis been performed appropriately and rigorously?

Reviewer #1: No

Reviewer #2: Yes

3. Have the authors made all data underlying the findings in their manuscript fully available?

Reviewer #1: Yes

Reviewer #2: Yes

4. Is the manuscript presented in an intelligible fashion and written in standard English?

Reviewer #1: Yes

Reviewer #2: Yes

**Reviewer #1: ** 5. Please add the age and number of rats in the animals’ subheading of the methods section.

6. Please write in vitro in an Italic format in whole parts of the manuscript.

7. Did you check the normality of data? What was the relevant test/s? Please add its detail in the statistical analysis.

8. In the results section, for describing statistical results, please provide details of the statistical test used, and full statistical reporting of the results. Full statistical reporting should include the number of observations, the test statistics (eg t, f values), exact probabilities and degrees of freedom for each of the observations where statistics are provided.

9. The quality of provided Figure 3 is not good and it’s blur. Please provide a better image. In addition, it’s not clear which parts of the brain were evaluated in the figure. For better clarification, please also add an image with lower magnification.

**Reviewer #2: ** Nizinska et al. aimed to investigate the role of Mbd3 (an epigenetic regulator in neuron) in the epileptic animal and cell models by overexpression of this protein. They found that overexpression of Mbd3 could decrease anxiety, increase excitability and accelerate epileptogenesis in the PTZ-kindling rats. And then they profiled the mRNA transcripts in cultured cells with magnesium deficiency-induced epileptiform discharges and identified complex and orchestrated time- and state-specific changes of a subset of genes which are involved in various pathways implicating in contribution to epileptogenesis. This work has potential values in the field of epilepsy research. However, I have some concerns as follows:

1) The anayses of transcriptome data seem to be preliminary. Additional bioinformatic analyses help to highlight which enriched functional terms may be more associated with overexpression of Mbd3.

2) Represented genes on transcriptic data should be analysed by qPCR both in vitro and in vivo.

3) Further, the CpG status of some represented genes may also be detected under epileptic and Mbd3 overexpression conditions, which can provide strong evidence for the role of Mbd3 in epileptogenesis.

4) Fig 1E, F and G: Related Western blot images should be present in this figure.

5) Some typo errors need to be revised.

**Do you want your identity to be public for this peer review?** For information about this choice, including consent withdrawal, please see our Privacy Policy

Reviewer #1: No

Reviewer #2: No

---

## [Author Response · Author response to Decision Letter 1]

11 Aug 2025

POINT-BY-POINT RESPONSE TO ALL REVIEWERS' COMMENTS

Reviewer #1:

Comment: Please add the age and number of rats in the animals’ subheading of the methods section.

Answer: To address this comment, we extracted some information from the "Animal Surgery" chapter and moved it to a newly created "Animals" chapter, and added the required information to that chapter. The added sentence reads:

“Altogether, 220 animals were used; two animals died during PTZ kindling. The body weight at the start of the experiment was 280-320 g.”

Comment: Please write in vitro in an Italic format in whole parts of the manuscript.

Answer: We apologize for this error. It has been corrected throughout the manuscript.

Comment: Did you check the normality of data? What was the relevant test/s? Please add its detail in the statistical analysis.

Answer: Normality of data distribution was assessed using the Kolmogorov–Smirnov test. This information is now added to the chapter ‘Statistical analysis”

Comment: In the results section, for describing statistical results, please provide details of the statistical test used, and full statistical reporting of the results. Full statistical reporting should include the number of observations, the test statistics (eg t, f values), exact probabilities and degrees of freedom for each of the observations where statistics are provided.

Answer: To address this comment, we have improved the description of statistics in the “Materials and Methods” section and have rewritten the whole “Results” section to provide the requested information.

Comment: The quality of provided Figure 3 is not good and it’s blur. Please provide a better image. In addition, it’s not clear which parts of the brain were evaluated in the figure. For better clarification, please also add an image with lower magnification.

Answer: In response to this suggestion, we provide higher-quality images acquired using confocal microscopy. We have also added information on the localization of the presented neurons in the description of the Figure 3 and added relevant information to Materials and Methods section

Reviewer #2

Comments: 1) The anayses of transcriptome data seem to be preliminary. Additional bioinformatic analyses help to highlight which enriched functional terms may be more associated with overexpression of Mbd3.

In response to this comment we have substantially updated the manuscript. We have performed an additional functional enrichment analysis focusing on Gene Ontology Biological Process (GO BP) terms for differentially expressed genes at each analyzed time point. Additionally we have updated the analysis of Clusters, changing the analysis performed with Reactome to GO BP providing a more accessible for the reader view of biological processes potentially associated with Mbd3 overexpression. The results are now presented in the updated Table 1, with enriched GO BP terms for each time point. Furthermore, we have substantially revised and extended the Discussion section to elaborate on the biological roles of selected genes and pathways identified in the analysis, thereby providing a more comprehensive interpretation of the transcriptomic changes observed.

Comments: 2) Represented genes on transcriptic data should be analysed by qPCR both in vitro and in vivo.

3) Further, the CpG status of some represented genes may also be detected under epileptic and Mbd3 overexpression conditions, which can provide strong evidence for the role of Mbd3 in epileptogenesis.

Answer: We fully agree that validating transcriptomic data by qPCR both in vitro and in vivo, as well as investigating the CpG methylation status of selected genes under epileptic conditions and Mbd3 overexpression, would provide critical mechanistic insights into the role of Mbd3 in epileptogenesis. At this stage, we are actively seeking funding to pursue these directions, which we intend to address in a follow-up study and subsequent manuscript.

Comment: 4) Fig 1E, F and G: Related Western blot images should be present in this figure.

Answer: To address this comment, we have added to Figure 1 panel H, presenting the Western Blot image, which shows the only statistically significant difference between protein levels following PTZ injection, that is, the 4h time point in the entorhinal cortex/amygdala.

Due to the large number of samples analyzed, the individual time points were run on separate gels. As a result, assembling a single panel with the corresponding Western blot images would require splicing individual lanes from different blots. We believe that such a figure would not be informative and could be potentially misleading. For this reason, we decided to show only representative images of the time point with a statistically significant difference.

Comment: 5) Some typo errors need to be revised.

Answer: We apologize for the typo errors. The text was checked for spelling, grammatical, and stylistic errors.

---

## [Decision Letter · Decision Letter 1]

27 Aug 2025

Dear Dr. Lukasiuk, 

Thank you for submitting your manuscript to PLOS ONE. After careful consideration, we feel that it has merit but does not fully meet PLOS ONE’s publication criteria as it currently stands. Therefore, we invite you to submit a revised version of the manuscript that addresses the points raised during the review process.

We look forward to receiving your revised manuscript.

Kind regards,

Giuseppe Biagini, MD

Academic Editor

PLOS ONE

Journal Requirements:

Reviewers' comments:

Reviewer's Responses to Questions

**Comments to the Author**

Reviewer #1: All comments have been addressed

Reviewer #2: (No Response)

2. Is the manuscript technically sound, and do the data support the conclusions?

Reviewer #1: Yes

Reviewer #2: (No Response)

3. Has the statistical analysis been performed appropriately and rigorously?

Reviewer #1: Yes

Reviewer #2: (No Response)

4. Have the authors made all data underlying the findings in their manuscript fully available?

Reviewer #1: Yes

Reviewer #2: (No Response)

5. Is the manuscript presented in an intelligible fashion and written in standard English?

Reviewer #1: Yes

Reviewer #2: (No Response)

Reviewer #1: All concerns have been fully addressed. As a minor issue, please rewrite the conclusion of the abstract.

Reviewer #2: The authors have addressed most of my concerns, and this manuscript has been improved upon revision.

**Do you want your identity to be public for this peer review?** For information about this choice, including consent withdrawal, please see our Privacy Policy

Reviewer #1: **Yes: ** Maryam Ghasemi-Kasman

Reviewer #2: No

---

## [Author Response · Author response to Decision Letter 2]

5 Sep 2025

Dear Prof Biagini and Reviewers,

Thank you for the opportunity to revise and resubmit our manuscript PONE-D-25-22943, entitled “The role of Methyl-CpG binding domain 3 (Mbd3) protein in epileptogenesis. We sincerely appreciate the time and effort that you and the Reviewers have invested in evaluating our work. We have carefully considered all the comments and suggestions, and we have revised the manuscript accordingly.

Below, we provide a detailed response to each point raised by the Reviewers.

POINT-BY-POINT RESPONSE TO ALL REVIEWERS' COMMENTS

Reviewer #1:

Comment: All concerns have been fully addressed. As a minor issue, please rewrite the conclusion of the abstract.

Answer: The abstract has been rewritten according to the Reviewer's suggestions and now reads:

Methyl CpG binding domain 3 (Mbd3) protein belongs to the MBD family of proteins and is responsible for reading the DNA methylation pattern. Our previous study revealed increased levels of Nucleosome Remodeling and Deacetylase (NuRD) complex proteins, including Mbd3, in the brains of epileptic animals. The present study investigated whether the Mbd3 protein level determines the seizure threshold.

We demonstrate that seizures induced by pentylenetetrazole (PTZ) cause a transient, brain area-specific increase in Mbd3 protein levels in the entorhinal cortex and amygdala. Overexpression of Mbd3 in the amygdala using AAV decreased anxiety, increased excitability in the open-field test, and accelerated epileptogenesis in the PTZ-kindling model. In vitro, mRNA profiling using RNA-seq in a model of magnesium deficiency-induced epileptiform discharges revealed complex, time- and state-specific changes in gene expression. Genes regulated by Mbd3 overexpression were associated with the Wnt and Notch pathways, potassium channel function, and GABAB receptor signaling.

Our findings indicate that increased Mbd3 expression has pro-epileptic properties and contributes to the regulation of multiple pathways potentially involved in seizure development. Significantly, seizures themselves transiently elevate Mbd3 levels, suggesting a potential vicious circle that may aggravate disease progression. Targeting the pro-epileptic effects of Mbd3 could therefore represent a novel therapeutic approach in epilepsy.

---

## [Editor Report · Decision Letter 2]

16 Sep 2025

The role of Methyl-CpG binding domain 3 (Mbd3) protein in epileptogenesis

PONE-D-25-22943R2

Dear Dr. Lukasiuk,

We’re pleased to inform you that your manuscript has been judged scientifically suitable for publication and will be formally accepted for publication once it meets all outstanding technical requirements.

Kind regards,

Giuseppe Biagini, MD

Academic Editor

PLOS ONE
---

## [Editor Report · Acceptance letter]

PONE-D-25-22943R2

PLOS ONE

Dear Dr. Lukasiuk,

I'm pleased to inform you that your manuscript has been deemed suitable for publication in PLOS ONE. Congratulations! Your manuscript is now being handed over to our production team.

Kind regards,

on behalf of

Dr. Giuseppe Biagini

Academic Editor

PLOS ONE